# Mild Dehydration Identification Using Machine Learning to Assess Autonomic Responses to Cognitive Stress

**DOI:** 10.3390/nu12010042

**Published:** 2019-12-23

**Authors:** Hugo F. Posada-Quintero, Natasa Reljin, Aurelie Moutran, Dimitrios Georgopalis, Elaine Choung-Hee Lee, Gabrielle E. W. Giersch, Douglas J. Casa, Ki H. Chon

**Affiliations:** 1Department of Biomedical Engineering, University of Connecticut, Storrs, CT 06269, USA; natasa.reljin@uconn.edu (N.R.); aurelie.moutran@uconn.edu (A.M.); dimitrios.georgopalis@uconn.edu (D.G.); ki.chon@uconn.edu (K.H.C.); 2Department of Kinesiology, Human Performance Laboratory, University of Connecticut, Storrs, CT 06269, USA; elaine.c.lee@uconn.edu (E.C.-H.L.); gabrielle.giersch@uconn.edu (G.E.W.G.); douglas.casa@uconn.edu (D.J.C.)

**Keywords:** dehydration, autonomic nervous system, electrodermal activity, pulse rate variability, machine learning

## Abstract

The feasibility of detecting mild dehydration by using autonomic responses to cognitive stress was studied. To induce cognitive stress, subjects (*n* = 17) performed the Stroop task, which comprised four minutes of rest and four minutes of test. Nine indices of autonomic control based on electrodermal activity (EDA) and pulse rate variability (PRV) were obtained during both the rest and test stages of the Stroop task. Measurements were taken on three consecutive days in which subjects were “wet” (not dehydrated) and “dry” (experiencing mild dehydration caused by fluid restriction). Nine approaches were tested for classification of “wet” and “dry” conditions: (1) linear (LDA) and (2) quadratic discriminant analysis (QDA), (3) logistic regression, (4) support vector machines (SVM) with cubic, (5) fine Gaussian kernel, (6) medium Gaussian kernel, (7) a k-nearest neighbor (KNN) classifier, (8) decision trees, and (9) subspace ensemble of KNN classifiers (SE-KNN). The classification models were tested for all possible combinations of the nine indices of autonomic nervous system control, and their performance was assessed by using leave-one-subject-out cross-validation. An overall accuracy of mild dehydration detection was 91.2% when using the cubic SE-KNN and indices obtained only at rest, and the accuracy was 91.2% when using the cubic SVM classifiers and indices obtained only at test. Accuracy was 86.8% when rest-to-test increments in the autonomic indices were used along with the KNN and QDA classifiers. In summary, measures of autonomic function based on EDA and PRV are suitable for detecting mild dehydration and could potentially be used for the noninvasive testing of dehydration.

## 1. Introduction

Clinicians and researchers need accurate, precise, relatively non-invasive, and sensitive measurements of hydration state, which is important to health and performance. The effects of dehydration are well-studied for their clinical implications [1,2]. Though the effects of mild dehydration equivalent to ≤2% body mass loss that are attributable to acute water losses are less understood, research has demonstrated that mild dehydration causes headache, tiredness, reduced alertness and cognitive performance, and greater difficulty concentrating [3]. Several studies have suggested that mild dehydration modifies the cerebrovascular response and the autonomic function in response to physical and cognitive stimuli [4,5,6], and it is likely that many other effects on health and performance have not yet been identified.

The available tools for assessing the degree of a person’s hydration state include body mass change, serum and urine osmolality, urine specific gravity, and urine volume [7,8,9,10,11]. However, no hydration index has been shown to be valid (or feasible) in all dehydration scenarios (laboratory and field) [7,12,13]. For this reason, noninvasive objective biomarkers for detecting mild dehydration that do not require laboratory testing are of special interest. Autonomic nervous system (ANS) variables are novel targets for the development of an objective physiological tool for detecting dehydration, given the mechanistic physiology of hydration and fluid homeostasis.

The physiological response to dehydration is an integrated and complex network including psychological and behavioral (e.g., thirst), cellular (e.g., cell volume regulation), gene expression (e.g., organic osmolyte production), and organ-level (e.g., cardiovascular) changes that attempt to regulate cell and blood plasma volume. During dehydration, nervous system chemoreceptors and mechanoreceptors detect variables like osmolality and concentrations of certain solutes to stimulate the release of fluid regulatory hormones [14]. Fluid-regulatory hormones, including arginine vasopressin (AVP, and antidiuretic hormone), renin, angiotensin II, aldosterone, and atrial natriuretic peptide, coordinate signaling to conserve body water, stimulate thirst, and protect cell and blood plasma volume. Fluid-regulatory hormones directly target renal and cardiovascular tissue and have direct action on vasoconstriction and dilation. It is likely that mild dehydration affects aspects of cardiovascular and ANS physiology that have yet to be studied and detected with sensitive tools.

The purpose of this study was to evaluate the feasibility of using autonomic responses to cognitive stress for the assessment of mild dehydration. A previous study attempted to assess dehydration that was produced by physical activity through the use of parameters that were obtained from heart rate, electrodermal activity (EDA), skin temperature, and body mass index by using an empirical formula [15]. The protocol only achieved a fluid loss of 0.53 in average, and the regression model fitted to predict fluid loss was not validated. In a recent study, we obtained an accuracy of 67.91% when detecting dehydration in patients with a dehydrating illness with a support vector machine (SVM) model based on the spectral features extracted from photoplethysmographic signals [16]. In this study, we conducted the validation of a model based on the indices of the ANS based on EDA and pulse rate variability (PRV). This a novel study that examines the feasibility of detecting mild dehydration by using measures of ANS based on EDA and PRV.

## 2. Materials and Methods

### 2.1. Protocol

Table 1 shows the study design for this experiment. Seventeen men (age: 23 ± 3.4 years old; weight: 81 ± 7.3 kg; height: 176.6 ± 5.6 cm) completed three consecutive morning experimental visits (days 1–3) for 30–40 min per visit. Participants were instructed to abstain from exercise and alcohol consumption throughout the study and to avoid altering normal sleep each night. Before participating, all subjects read and signed informed consent documents for inclusion, previously approved by the University of Connecticut’s Institutional Review Board (protocol H17-291).

During all days of this study, all urine produced (during each 24 h period) was collected in a jug that was provided by the investigators. Subjects also received a cloth bag for carrying this urine jug during all daily activities. During each of the visits to the experimental lab, participants provided a blood sample (2 tablespoons = 29.57 mL) from a forearm vein and a small urine sample.

On the morning of day 1, baseline information (body weight, height, age, perceptual ratings, blood sample, and urine sample) was taken. Participants were then instructed to eat and drink normally (euhydration) during their daily activities on this first day. Fluid restriction began on day 2. During this day, besides the total fluid restriction, we provided a list of dry foods and instructed the participants not to eat watery foods (e.g., soup, watermelon, oranges, grapes, smoothies, and milk). On day 3, participants were expected to be mildly dehydrated. Two sets of measurements and tests were conducted on this day. The first set of measurements was conducted upon arrival to the experimental facility (in dehydration). After this set of measurements was completed, participants were instructed to drink as much water as desired for 30 min before the second set of measurements of the day (Measure 4 overall). After participants’ rehydration, they were instructed to return to normal fluid and food intake.

### 2.2. Stroop Task

In order to induce cognitive stress, subjects performed the Stroop task every visit. In this task, subjects were asked to stay at rest for 4 min (defined as the rest stage). After that, the subjects performed a test in which they had to say the color of words that appeared on the screen of a tablet computer; the words named colors [17] (defined as the test stage). The screen showed congruent visualizations (the word was written in the color it expressed) and incongruent visualizations (the word and the color it was printed in were different to induce cognitive stress). The words and colors were: “blue,” “yellow,” “green,” “red,” “purple,” and “black.” The color of the background of the screen also changed to be randomly congruently or incongruently colored with the name of the word, never matching the color the word was printed in. The test stage lasted 4 min. On the morning of day 3, participants completed the Stroop task (comprising the rest and test stages) two times. The first test was administered before consuming any water, and the second was administered following the 30-min period of unlimited access to water.

On all days, EDA and photoplethysmographic (PPG) signals were collected during the rest and test stages of the Stroop task. EDA was collected with the use of a pair of stainless-steel electrodes placed in the left hand and a galvanic skin response module FE116 (ADInstruments, Sydney, Australia). PPG signals were collected with the use of a wearable device on the left wrist (Samsung Simband). The sampling frequency was set to 32 Hz for both signals.

### 2.3. Data Processing

Measures to assess the autonomic nervous system based on EDA and pulse rate variability (PRV) were computed by using the data collected. To ensure high-quality physiological data, subjects were asked to keep their left hands still—as this hand was where the data were collected—while performing the Stroop task. A summary of the indices of EDA and PRV computed in this study is included in Table 2.

#### 2.3.1. Indices of PRV

In this study, we used PRV extracted from PPG signals. PRV is a suitable alternative measurement of heart rate variability [18,19,20], with the advantage that photoplethysmography is a low-cost optical technique [21] that is much simpler to use than the electrocardiogram and is easily deployed in smartphones and wearable devices [22,23,24]. For PRV analysis, four minutes of clean PPG signals were extracted from the rest and test stages of the Stroop task. A customized algorithm was used for PPG peak detection. Though PPG signal morphology is highly individual-dependent, we did not use the information from the morphology; instead, we only used the peak-to-peak (also called pulse-to-pulse) interval variability. In other words, we only obtained the variations in the time between pulses of the PPG signals, from which PRV analysis was made. To ensure data quality, all segments were visually inspected to make sure that no beat was missed. The peak-to-peak interval series was converted to an evenly time-sampled signal (4 Hz) through cubic spline interpolation. A Blackman window (length of 256 points) was applied to each segment, and the fast Fourier transform was calculated for each windowed segment. Finally, the power spectra of the segments were averaged.

The index of low frequencies of PRV (PRVLF [ms2], 0.045–0.15 Hz), the index of high frequencies of PRV (PRVHF [ms2], 0.15 to 0.4 Hz), and their normalized versions (PRVLFn and PRVHFn, obtained by dividing by total power of PRV in normalized units [n.u.]) were computed [25]. Indices from the LF range (PRVLF and PRVLFn) of PRV are referenced as indices of sympathetic control, and indices from the HF power (PRVHF and PRVHFn) are commonly used as indices of parasympathetic control.

#### 2.3.2. Indices of Electrodermal Activity

In this study, the first two minutes of EDA data were extracted from both the rest and test stages of the Stroop task to compute the indices of EDA in the time and frequency domains. In the time domain, the skin conductance level (SCL, expressed in microsiemens, µS) and the skin conductance responses (SCRs) were obtained [26]. SCL is defined as the mean value of the tonic component of the EDA, and the SCRs are the phasic changes of the EDA signal. Based on the detected SCRs, the frequency of non-specific SCRs (NS.SCRs) was computed as the number of SCRs whose amplitude was higher than a given threshold (0.05 µS in this study) per minute [26]. The SCL and NS.SCRs indices were extracted by using nonnegative sparse deconvolution for the decomposition of EDA into tonic and phasic components [27].

The power spectral index of EDA, EDASymp [µS2], was computed by integrating the power in the range of 0.045–0.25 Hz, as it was previously found to be sensitive to cognitive stress [28]. The spectra of EDA were calculated by using Welch’s periodogram method with a 50% data overlap. A Blackman window (length of 128 points) was applied to each segment, the fast Fourier transform was calculated for each windowed segment, and the power spectra of the segments were averaged. To compute the time-varying index of EDA, TVSymp, the time-frequency representation of EDA was computed by using variable frequency complex demodulation (VFCDM), a time-frequency spectral analysis technique that provides accurate amplitude estimates and one of the highest time-frequency resolutions [29]. At a sampling frequency of VFCDM decomposition of 2 Hz, the second and third components, comprising the approximate range of 0.08–0.24 Hz, were used to compute TVSymp, as defined in a previous study [30].

We evaluated the increments between the rest and test (cognitive stress) stages of the Stroop task in the indices of EDA and PRV for the four measurements. The normality of each index was tested by using the one-sample Kolmogorov–Smirnov test [31,32,33]. To test the significance of rest-to-test increments, we used the *t*-test for normally-distributed indices, and we used the two-sided Wilcoxon rank sum test if non-normality was found [34]. A *p* value of <0.05 was considered significant.

### 2.4. Classification Analysis

The physiological indices of the ANS collected in this study were: PRVLF, PRVLFn, PRVHF, PRVHFn, SCL, NSSCRs, EDASymp, EDASympn, and TVSymp. Measurements of days 1, 2, and 3b (after the subject drank as much water as desired), were defined as the samples of the “wet” (no dehydration) class. Measurements on day 3a were considered part of the “dry” (mild dehydration) class.

Nine approaches were tested for the classification of “wet” and “dry” conditions (Table 3): discriminant analysis with linear and quadratic discriminant functions [35,36]; logistic regression (LR) [37]; support vector machines (SVM) with cubic, fine Gaussian (C = 1, γ = 0.66), and medium Gaussian (C = 1, γ = 2.6) kernel functions [38]; decision trees (DT) [39]; k-nearest neighbor classifier (KNN, k = 1) [40,41]; and a subspace ensemble of KNN classifiers (SE-KNN; 30 learning cycles) [42,43,44]. In this study, we use the terms “fine” and “medium” to differentiate the two classifiers based on the value of γ. The models were trained by using only rest measures, only test measures, increments (test measure minus rest measure, denoted as rest-to-test increments), and both measures (test and rest). By doing this, we evaluated whether the indices of EDA and PRV could detect mild dehydration in a purely static test (i.e., one that requires only a measure at rest) and how a dynamic test (i.e., a test that requires a measure at rest and a measure in response to a test) improves detection.

To prevent overfitting, leave-one-subject-out cross-validation was used to evaluate the performance of the constructed models. The data consisted of 51 (3 times 17 subjects) “wet” samples and 17 “dry” samples. The three “wet” samples corresponded to the baseline measurement (Measure 1, taken on day 1 before the euhydration day), the measurement after the euhydration day (Measure 2, taken on day 2, before the fluid restriction day), and the measurement after rehydration (Measure 4, taken on day 3, after the fluid restriction day). Leaving one subject out for the validation resulted in the use of 48 “wet” samples and 16 “dry” samples in each training set. In order to obtain balanced classes, the “dry” samples were up-sampled in the training process. Given the structure of the dataset (“wet” samples were exactly three times the “dry” samples), this was equivalent to giving a higher weight to the “dry” class than “wet” class. Every leave-one-subject-out testing set was comprised of three “wet” samples and one “dry” sample. Accuracy ((correctly classified as “dry” + correctly classified as “wet”)/total) was used as the main measure of model performance. The error rate ((incorrectly classified as “dry” + incorrectly classified as “wet”)/total), the false positive rate (incorrectly classified as “dry”/actual “dry”), specificity (correctly classified as “wet”/actual “wet”), and precision (correctly classified as “dry”/all classified “dry”) were also computed.

The indices (i.e., features) of the data set were scaled to improve the performance of the machine learning algorithms by using normalization to the Euclidean norm of each index. The classification analysis was performed in MATLAB (MathWorks, Inc., Natick, MA 01760, USA).

## 3. Results

### 3.1. Dehydration Assessment

Table 4 includes the results for urinary loss, blood osmolality, body mass, and body-mass loss encountered in the baseline measurement (Measure 1; taken on day 1), after the euhydration day (Measure 2; taken on day 2), after the fluid restriction day (Measure 3, taken on day 3), and after rehydration (Measure 4, taken on day 3). Urinary losses amounted to 0.83 ± 0.28 L over 24 h of fluid restriction, a value that was significantly lower than that during the euhydration day. Blood osmolality increased significantly during the 24 h fluid restriction day compared with the corresponding time point after the euhydration day. Overall, subjects’ body mass after the euhydration day was 81.1 ± 0.48 kg, and it was 79.7 ± 7.1 kg after 24 h of fluid restriction. This equates to 1.78 ± 0.48% loss of body mass during the fluid restriction day and indicates that the protocol successfully induced mild dehydration.

### 3.2. Indices of Autonomic Response

Figure 1 shows the EDA and heart rate (HR) data for Measures 1–4 for a given subject. Table 5 includes the results for the rest and test stages of the Stroop task measurements over three days of tests (four measurements in total). SCL, NS.SCRs, EDASympn, TVSymp, PRVLFn and PRVHFn were normally distributed. EDASymp, PRVLF and PRVHF were non-normally distributed. Test vs. rest significant differences of the indices are marked in the table. The only index that exhibited consistently significant differences was TVSymp. The other indices exhibited variability in their rest-to-test increments.

### 3.3. Classification Analysis

We used the minimum redundancy maximum relevance [45] (MRMR) and joint mutual information [46,47] (JMI) approaches for feature selection. However, MRMR and JMI feature selection achieved consistent good accuracy for all the data types classifications (e.g., only rest and only test). For this reason, the classification models were trained for all possible combinations of the nine indices of the ANS (512 combinations). Table 6 includes the results for the most accurate classification approaches for each dataset used.

Only the combination of indices and machine learning models that achieved the highest accuracy for the four possible data types (i.e., only rest, only test, rest-to-test increments, both test and rest measures) were included. When only rest measures were used (e.g., data from the test stage of the Stroop task were not used), SE-KNN achieved an accuracy of 91.2% by using EDASymp, TVSymp and PRVHFn in classifying “dry”/“wet” classes (Figure 2 shows the scatterplot for this specific combination of indices). Note that the dataset looks imbalanced in Figure 2 because the up-sampled “dry” samples fell on top of the other “dry” samples. The same “dry”/“wet” classification accuracy (91.2%) was achieved by using the Cubic SVM classifier with the only test values (i.e., no data were collected during the rest stage of the Stroop task) of the NS.SCRs, EDASymp, EDASympn, PRVLF, PRVLFn and PRVHFn indices. This cubic SVM model was the best in overall, as its error rate was the lowest (8.8%), its sensitivity (100%) was the highest, and its specificity (88.2%) and precision (73.9%) were high. A slightly lower “dry”/“wet” classification accuracy was achieved when rest-to-test increments (i.e., difference between PRV and EDA values from the rest-to-test stages of the Stroop task) were used in the KNN model (86.8%) with SCL, NS.SCRs, TVSymp, PRVLF, PRVLFn and PRVHFn. An accuracy of 86.8% was reached when both rest and test measures were used in a quadratic discriminant analysis (QDA) classifier. Remarkably, this model exhibited a specificity of 98% and a precision of 90%, the highest overall. This means this model detected the “wet” samples almost unequivocally. However, it had a rather low sensitivity (52.9%).

## 4. Discussion

In this study, we collected EDA and PRV data from healthy subjects who performed the Stroop task, which consisted of a stage of rest followed by a stage of test, at intervals throughout the course of a standardized experiment that caused mild dehydration. We collected baseline data on the first day, then took a measure following a day of normal hydration (euhydration), another measure after a day of fluid restriction, and a final measure on that same day but after rehydration. We defined the “dry” class as the measure taken right after the day of fluid restriction and the “wet” class as all the other measures (baseline, euhydration, and rehydration). Our subjects’ average body-mass loss of 1.78 ± 0.48% resulting from 24 h of fluid restriction was similar to that observed in previous studies [3,48,49,50]. This body-mass loss corresponded to the “dry” measures. There was no significant body-mass loss in the “wet” measures. We used several machine learning techniques to perform the “dry”/“wet” classification. Nine indices of autonomic control were obtained from the EDA and PRV data taken during the rest and test stages of the Stroop task. We trained the classifiers by using different combinations of the indices that were computed during the rest and test stages in order to evaluate the feasibility of the “dry”/“wet” classification in each case: (1) only measures taken at rest, (2) only measures taken during the Stroop task test, (3) the difference between rest and test measures (rest-to-test increments in EDA values), and (4) both rest and test measures.

We have demonstrated that mild dehydration can be accurately detected by measuring the effects of cognitive stress on the autonomic reactions expressed in EDA and PRV. By using a static approach (without Stroop task stimuli), in which measures are taken only when subjects are at rest, mild dehydration can be detected with an overall accuracy of 91.2%. If the dynamics of the autonomic response to cognitive stress are considered, mild dehydration can be detected with the same accuracy of 91.2% (using measures only under cognitive stress) and 86.8% (if the rest-to-test increments are used for the model). As for the measures of the ANS, the only indices that exhibited significant differences between the rest and test stages across the four measurements were the SCL and TVSymp. NS.SCRs exhibited significant differences for only three measurements, not including Measure 2 (day 2), and PRVLFn was only different in the third measure (first measure of day 3). This is in agreement with a recent study that established that measures of sympathetic arousal based on EDA (mainly TVSymp) are more reproducible in response to cognitive stress [51]. Furthermore, this study has shown that SCL and TVSymp are robust indices of cognitive stress in the presence of mild dehydration.

SE-KNN and Cubic SVM analysis were the most accurate classification tools in this study. This Cubic SVM model included the rest-to-test increments of NS.SCRs, EDASymp, EDASympn, PRVLF, PRVLFn, and PRVHFn. This suggests that both EDA and PRV bear information of the autonomic response that is necessary for detecting mild dehydration under cognitive stress. The SE-KNN classification model based on static measures included the EDASymp, TVSymp and PRVHFn, which indicates that in resting conditions, peripheral sympathetic reactions (expressed in EDA) better discriminate between normal and dehydrated subjects. This static model has a reduced sensitivity (76.5%) as compared to the Cubic SVM model (100%).

Models based on KNN and QDA that contained the only test and both the rest and test measurements were highly accurate (86.8% for both) and specific (86.3% and 98%, respectively). These two models used SCL, NS.SCRs, PRVLF, and PRVLFn, and they only differed in the two indices used in the KNN that were not used in the QDA (TVSymp and PRVHFn). All four models were highly specific. This means that a “dry” adjudication in these models is useful for detecting dehydration, because they rarely adjudicate a “dry” sample to a “wet” sample. A model with high specificity will accurately exclude dehydration from normally hydrated subjects, and a “dry” adjudication signifies a high probability of dehydration.

As for the limitations of the study, given the procedural restrictions, this study was only conducted on male subjects. The validity of the results for female subjects needs be tested in the future. Furthermore, this experiment was meant to evaluate the feasibility of EDA and PRV to detect mild dehydration in a controlled environment that afforded pre- and post-dehydration measurements. A more realistic situation in which no repeated measurements are taken should be tested in the future. However, our current results already show differences between “dry” and “wet” samples without repeated measurements. For example, day 2 measurements are not needed if we are solely interested in detecting differences between “wet” and “dry” states. EDA and PRV can be also affected by other confounders, like physiological stress and external stimuli. The feasibility of detecting dehydration in a more realistic setup should also be tested in the future.

## 5. Conclusions

In summary, measures of autonomic function based on EDA and PRV are suitable for discriminating mildly dehydrated subjects and have the potential to be used in a noninvasive and easy-to-deploy test of dehydration because EDA and PPG signals can be collected with the use of wearable devices. The most accurate models based only on test measurements evenly included indices of EDA and PRV, but the most accurate classification model based on static measures mainly relied on spectral indices of EDA. The practical application of our findings is the potential for development of wearable devices that use these variables to monitor the real-time hydration state. as validated with current validated and well-researched hydration biomarkers.

## Figures and Tables

**Figure 1 nutrients-12-00042-f001:**
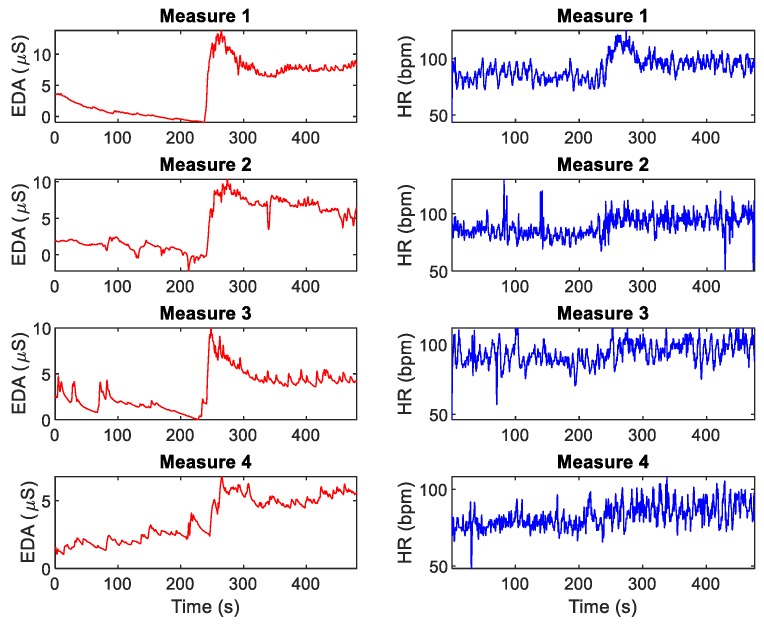
EDA and heart rate (HR) during rest and test stages of the Stroop task, for a given subject.

**Figure 2 nutrients-12-00042-f002:**
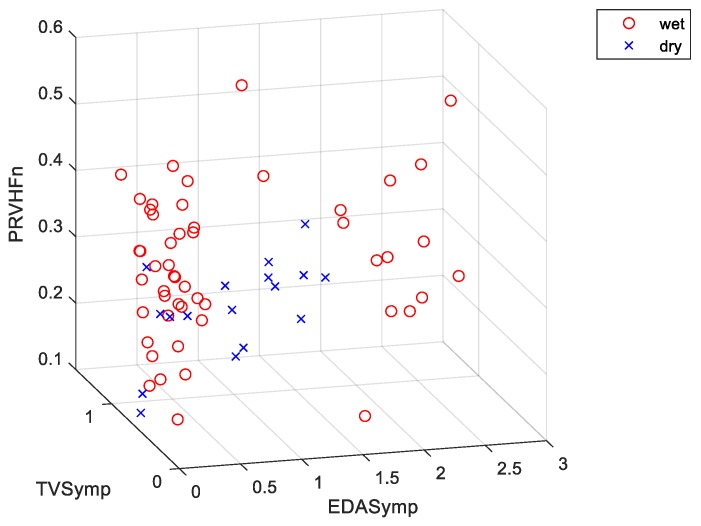
Scatter plot for the indices used in one of the most accurate models (accuracy = 91.2%). Classifier: subspace ensemble of k-nearest neighbor (KNN) classifiers (SE-KNN). The features are the rest measurements of: EDASymp (power spectral index of EDA), TVSymp (time-varying index of EDA), and PRVHFn (normalized high frequencies of PRV).

**Table 1 nutrients-12-00042-t001:** Study design.

	Day 1	Day 2	Day 3
Measures taken	Measure 1	Measure 2	Measure 3, Measure 4
Activities	Euhydration. Drink and eat normally	Fluid restriction. Drink no fluids and eat dry food	After Measure 3, consume water as desired

**Table 2 nutrients-12-00042-t002:** Indices of autonomic response based on electrodermal activity (EDA) and pulse rate variability (PRV).

		Name	Description
EDA	SCL	Skin conductance level	Mean value of the tonic component
	NS.SCRs	Non-specific SCRs	Frequency of phasic drivers >0.05 µS
	EDASymp	Power spectral index of EDA	Power of EDA in the range of 0.045–0.25 Hz
	EDASympn	Normalized EDASymp	EDASymp normalized to total power of EDA
	TVSymp	Time-varying index of EDA	Instantaneous amplitude of sympathetic components
PRV	PRVLF	Low frequencies of PRV	Power in the range of 0.045–0.15 Hz
	PRVLFn	Normalized PRVLF	PRVLF normalized to total power of PRV
	PRVHF	High frequencies of PRV	Power in the range of 0.15–0.4 Hz
	PRVHFn	Normalized PRVHF	PRVHF normalized to total power of PRV

**Table 3 nutrients-12-00042-t003:** Classification models.

Name	Description
LDA	Linear discriminant analysis
QDA	Quadratic Discriminant analysis
Logistic	Logistic regression model
Cubic SVM	SVM with cubic kernel
Fine Gaussian SVM	SVM with Gaussian kernel, C = 1, γ = 0.66
Medium Gaussian SVM	SVM with Gaussian kernel, C = 1, γ = 2.6
KNN	k-nearest neighbor classifier
DT	Decision trees
SE-KNN	Subspace ensemble of KNN classifiers

**Table 4 nutrients-12-00042-t004:** Body mass, body-mass loss, urinary loss and blood osmolarity measurements during the study.

	Day 1 (BL) Measure 1	Day 2 (EU) Measure 2	Day 3 (FR) Measure 3	Day 3 (RH) Measure 4
Urinary loss (liters)	-	1.9 ± 1.1	0.83 ± 0.28 *	-
Blood osmolality	277.8 ± 10.8	275.8 ± 21.9	286.6 ± 7.2 *	281.6 ± 9.7 *
Body-mass (kg)	81 ± 7.3	81.1 ± 7.2	79.7 ± 7.1 *	80.9 ± 7.3 *
Body-mass loss (%)	-	0.1 ± 0.8	−1.78 ± 0.48	−0.31 ± 0.66

* denotes significant difference (*p* < 0.05); BL: baseline; EU: euhydration; FR: fluid restriction; and RH: rehydration.

**Table 5 nutrients-12-00042-t005:** Measurements of autonomic response based on EDA and PRV during the rest and test stages of the Stroop task throughout the study.

	Day 1 (BL) Measure 1		Day 2 (EU) Measure 2		Day 3 (FR) Measure 3		Day 3 (RH) Measure 4	
	Rest	Test	Rest	Test	Rest	Test	Rest	Test
SCL (µS)	2.2 ± 2.4	7.1 ± 4.4 *	2.3 ± 3.6	5.5 ± 6.4 *	3.8 ± 4.7	7.9 ± 6.4 *	3.8 ± 5.5	6.3 ± 6.6 *
NS.SCRs (#/min)	3.8 ± 1.9	8.1 ± 2.1 *	4.2 ± 2.6	6.6 ± 3.8	3.9 ± 2.5	8.2 ± 3 *	3.9 ± 2.2	7.3 ± 3.6 *
EDASymp (µS^2^)	0.2 ± 0.48	6.6 ± 24 *	0.19 ± 0.33	0.97 ± 3.1	1.3 ± 4.8	0.61 ± 1.1	0.092 ± 0.17	0.096 ± 0.086
EDASympn (n.u.)	0.36 ± 0.21	0.27 ± 0.21	0.35 ± 0.18	0.24 ± 0.21	0.3 ± 0.17	0.2 ± 0.16	0.36 ± 0.2	0.27 ± 0.15
TVSymp (dimensionless)	0.52 ± 0.31	1.5 ± 0.42 *	0.69 ± 0.35	1.2 ± 0.48 *	0.55 ± 0.35	1.3 ± 0.44 *	0.72 ± 0.43	1.2 ± 0.48 *
PRVLF (mS^2^)	14 ± 12	15 ± 12	13 ± 15	15 ± 15	170 ± 650	120 ± 420	13 ± 12	19 ± 28
PRVLFn (n.u.)	0.35 ± 0.16	0.3 ± 0.14	0.29 ± 0.11	0.35 ± 0.14	0.34 ± 0.18	0.46 ± 0.15 *	0.32 ± 0.12	0.39 ± 0.18
PRVHF (mS^2^)	15 ± 23	16 ± 18	12 ± 10	17 ± 22	37 ± 110	55 ± 140	17 ± 22	27 ± 69
PRVHFn (n.u.)	0.29 ± 0.17	0.26 ± 0.13	0.34 ± 0.19	0.3 ± 0.13	0.29 ± 0.18	0.25 ± 0.14	0.35 ± 0.17	0.29 ± 0.13

* denotes significant difference to rest (*p* < 0.05). # represents the number of SCRs whose amplitude was higher than 0.05 µS. BL: baseline; EU: euhydration; FR: fluid restriction; RH: rehydration; SCL: skin conductance level; NS.SCRs: nonspecific skin conductance responses; EDASymp: sympathetic component of the EDA; TVSymp: time-varying index of sympathetic tone; PRVLF: low-frequency components of pulse rate variability (PRV); and PRVLFn: normalized low-frequency components of PRV.

**Table 6 nutrients-12-00042-t006:** Classification results for the most accurate models for each case.

Data Used	Classifier	Indices	Accuracy	Error Rate	Sensitivity	FPR	Specificity	Precision
Only rest	SE-KNN	EDASymp, TVSymp, PRVHFn	91.2%	8.8%	76.5%	3.9%	96.1%	86.7%
Only test	Cubic SVM	NS.SCRs, EDASymp, EDASympn, PRVLF, PRVLFn, PRVHFn	91.2%	8.8%	100.0%	11.8%	88.2%	73.9%
(Test-rest)	KNN	SCL, NS.SCRs, TVSymp, PRVLF, PRVLFn, PRVHFn	86.8%	13.2%	88.2%	13.7%	86.3%	68.2%
Rest and test	QDA	SCL, NS.SCRs, PRVLF, PRVLFn	86.80%	13.2%	52.9%	2.0%	98.0%	90.0%

SCL: skin conductance level; NS.SCRs: nonspecific skin conductance responses; EDASymp: sympathetic component of the EDA; TVSymp: time-varying index of sympathetic tone; PRVLF: low-frequency components of pulse rate variability (PRV); and PRVLFn: normalized low-frequency components of PRV.

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
