# Peer review of "Mild Dehydration Identification Using Machine Learning to Assess Autonomic Responses to Cognitive Stress"

_nutrients, 2019, doi:10.3390/nu12010042_

Round 1
Reviewer 1 Report
The authors have adequately responded to this reviewers concerns.
Reviewer 2 Report
The revised manuscript has been improved over the original version and addressed my earlier concerns.
This manuscript is a resubmission of an earlier submission. The following is a list of the peer review reports and author responses from that submission.
Round 1
Reviewer 1 Report
An interesting and well-executed study that may lead to novel non-invasive methodology for the detection of mild dehydration.
The manuscript is too a large extent well-written with a clear overview of existing literature in the field, and a concise description of the methodology.
What was not clear to me is whether the stroop test provides any additional advantage over the static approach with measurements during the resting state alone. Similarly, it is not clear to this reviewer whether one single measurement for the "wet" state would be sufficient to obtain adequate accuracy of the prediction. A test that requires three measurements in the "wet" state plus one measurement in the "dry" state over three or four days, would be quite cumbersome and demanding for practical use. This should be better presented in the results and explained in the discussion.
Under limitations it should be mentioned that the study was only assessed in males but not females.
Finally, please provide an internationally acknowledged unit (mL) for the measurement of blood volume taken.
Author Response
We want to thank the reviewer for the comprehensive revision of our manuscript. We have included a list of updates we have made to the manuscript, considering your comments and suggestions. Responses and changes made to the manuscript are highlighted.
Reviewer 1
An interesting and well-executed study that may lead to novel non-invasive methodology for the detection of mild dehydration.
The manuscript is too a large extent well-written with a clear overview of existing literature in the field, and a concise description of the methodology.
Comment:
What was not clear to me is whether the stroop test provides any additional advantage over the static approach with measurements during the resting state alone.
Response:
Thank you for your question. The EDA is thought to be elevated only by sympathetic reaction, like that elicited by cognitive stress. We included the Stroop task to procure cognitive stress and observe if EDA elevation in the presence of cognitive stress was affected by dehydration. However, it would be much more practical if the EDA could detect dehydration without a cognitive test. For that reason, we tested both the Stoop task and the static (without Stroop) approach. The first row of Table 6 shows this result. The first row labelled “only rest” compares the Day 3 (wet) rest state to other dry rest states (Days 1, 2, and 3). As shown in Table 6, the accuracy is 91.2% in differentiating the wet state vs. dry state without the induction of the Stroop task. Thus, the EDA, in conjunction with PRV, can provide sensitive detection of dehydration without the need for a cognitive task.
Comment:
Similarly, it is not clear to this reviewer whether one single measurement for the "wet" state would be sufficient to obtain adequate accuracy of the prediction. A test that requires three measurements in the "wet" state plus one measurement in the "dry" state over three or four days, would be quite cumbersome and demanding for practical use. This should be better presented in the results and explained in the discussion.
Response:
Thank you for your comment. This experiment was meant to evaluate the feasibility of EDA and PRV to detect mild dehydration in a controlled environment. In a future study, we will evaluate a more realistic situation. For example, a study in which control and dehydrated subjects will be tested, without repeated measurements. However, our current results already show differences between dry and wet without repeated measurements. For example, only day 2 measurements are needed if we are solely interest in detecting differences between wet and dry states.
Actual changes implemented:
This text has been added to the limitations paragraph.
“Furthermore, this experiment was meant to evaluate the feasibility of EDA and PRV to detect mild dehydration in a controlled environment that required pre and post-dehydration measurements. A more realistic situation in which no repeated measurements are required should be tested in the future. However, our current results already show differences between dry and wet without repeated measurements. For example, day 2 measurements are not needed if we are solely interest in detecting differences between wet and dry states.”
Comment:
Under limitations it should be mentioned that the study was only assessed in males but not females.
Response:
Thank you for your comment. We have added the gender restriction to the paragraph explaining the limitations of the study.
Actual changes implemented:
“As for the limitations of the study, given the procedural restrictions, this study was only conducted in male subjects. The validity of results in female subjects needs be tested in the future”
Comment:
Finally, please provide an internationally acknowledged unit (mL) for the measurement of blood volume taken.
Response:
Thank you for your suggestion. We have added the text as required.
Actual changes implemented:
“participants provided a blood sample (2 tablespoons = 29.57 mL) from a forearm vein and a small urine sample”
Reviewer 2 Report
This manuscript collects electrodermal activity (EDA) and pulse rate variability (PRV) signals, and uses machine learning algorithms to identify mild dehydration from those data. Please see detailed comments below:
The paper shows classification of EDA and PRV for dehydration. However, other physiological process could also lead to changes in EDA and PRV. How to rule that out in real application? There are 17 subjects in the study, but 51 wet samples, why? A number of 51 wet samples and 17 dry samples were used in classification, indicating an unbalanced dataset. How does that affect the classification? In this paper, many classifiers and combinations of features are attempted, and the some of the best ones were reported. Given the relatively small data size, could the obtained high accuracy just be a coincidence? In other words, the high classification performance doesn’t prove the effectiveness of the EDA and PRV features. The PPG signal is highly individual-dependent. It’s not clear why derivative features of PPG contain predictive information of mild hydration.
Author Response
We want to thank the reviewer for the thorough revision of our manuscript. A list of updates we have made to the manuscript in response to your comments and suggestions is included. Responses and changes made to the manuscript are highlighted.
Reviewer 2
This manuscript collects electrodermal activity (EDA) and pulse rate variability (PRV) signals, and uses machine learning algorithms to identify mild dehydration from those data. Please see detailed comments below:
Comment:
The paper shows classification of EDA and PRV for dehydration. However, other physiological process could also lead to changes in EDA and PRV. How to rule that out in real application?
Response:
Thank you for your astute comment. We acknowledge the possible interfering physiological processes that could lead to changes in EDA and PRV. For this reason, this experiment was conducted in a controlled environment, with subjects’ testing restricted to an experimental setup. The feasibility of detecting dehydration in a more realistic setup should be tested in the future.
Actual changes implemented:
We have added the following text to the limitations paragraph of the manuscript:
“EDA and PRV can be also affected by other confounders, like physiological stress and external stimuli. The feasibility of detecting dehydration in a more realistic setup should also be tested in the future.”
Comment:
There are 17 subjects in the study, but 51 wet samples, why?
Response:
Thank you for your question. As we performed three measurements while each subject was hydrated (Day 1: euhydration, Day 2: before fluid restriction, and Day 3: measurement 4 after rehydration), there are three times 17 wet samples. This has now been noted in line 187. We have added some more text to further clarify this.
Actual changes implemented:
“The three “wet” samples correspond to baseline measurement (measure 1, taken on day 1 before euhydration day), the measurement after the euhydration day (measure 2, taken on day 2, before fluid restriction day), and after rehydration (measure 4, taken on day 3, after fluid restriction day).”
Comment:
A number of 51 wet samples and 17 dry samples were used in classification, indicating an unbalanced dataset. How does that affect the classification?
Response:
Thank you for your comment. We took measures to avoid the undesirable effects of class imbalance. In order to obtain balanced classes, the “dry” samples were up-sampled in the training process. Given the structure of the dataset, this is equivalent to giving a higher weight to the “dry” class than wet class. Furthermore, as can be seen from the results, the classification was not affected by the imbalance of the data as the models’ performance was high in the leave-one-subject-out cross-validation. We have stressed this in the manuscript.
Actual changes implemented:
“In order to obtain balanced classes, the “dry” samples were up-sampled in the training process. Given the structure of the dataset (“wet” samples are exactly three times “dry” samples), this is equivalent to giving a higher weight to the “dry” class.”
Comment:
In this paper, many classifiers and combinations of features are attempted, and the some of the best ones were reported. Given the relatively small data size, could the obtained high accuracy just be a coincidence? In other words, the high classification performance doesn’t prove the effectiveness of the EDA and PRV features.
Response:
Thank you for your comment. The reviewer is correct that testing many combinations of classifiers and features could lead to coincidental good performance. However, that is true in the training process. The models (combination of machine learning algorithm and features) are evaluated using leave-one-subject-out cross-validation, which means the model is trained 17 times (number of features); each time the model is trained with all but one subject’s data, then tested on the subject’s data that was not used for training. This assures that the overall performance is not affected by over-fitting and reduces the risk that the classification performance is obtained by chance.
Comment:
The PPG signal is highly individual-dependent. It’s not clear why derivative features of PPG contain predictive information of mild hydration.
Response:
Thank you for your comment. Although the PPG signal morphology is highly individual-dependent, we are not using the information from the morphology, but only the peak-to-peak (also called pulse-to-pulse) interval variability. In other words, we only obtained the variations in the time between pulses of PPG, from which PRV analysis is deployed. The inter-subject variability of the PPG signal is not absorbed by the resulting PRV features. We have added this comment to the manuscript.
Actual changes implemented:
“Although the PPG signal morphology is highly individual-dependent, we did not use the information of the morphology, but only the peak-to-peak (also called pulse-to-pulse) intervals variability. In other words, we only obtained the variations in the time between pulses of PPG, from which PRV analysis is deployed.”